# Impact of vented and condenser tumble dryers on waterborne and airborne microfiber pollution

**Amber M. Cummins**[1], **Adam K. Malekpour**[2], **Andrew J. Smith**[2], **Suzanne Lonsdale**[1], **John R. Dean**[1], **Neil J. Lant**[2]*

**1** Department of Applied Sciences, Northumbria University, Newcastle upon Tyne, United Kingdom,
**2** Procter & Gamble, Newcastle Innovation Center, Newcastle upon Tyne, United Kingdom

* lant.n@pg.com

**Data Availability Statement:** All relevant data are within the manuscript and its Supporting Information files.

**Funding:** The study was entirely funded by the following three sources: The Worshipful Company

## Abstract

Laundering of textiles is a significant source of waterborne microfiber pollution, and solutions are now being sought to mitigate this issue including improvements in clothing technology and integration of filtration systems into washing machines. Vented tumble dryers are a potential source of airborne microfiber pollution, as their built-in lint filtration systems have been found to be inefficient with significant quantities of textile microfibers being released to the external environment through their exhaust air ducts. The present study is the first to evaluate the impact of condenser dryers, finding that they are significant contributors to waterborne microfiber pollution from the lint filter (if users clean this with water), the condenser and the condensed water. Microfiber release from drying of real consumer loads in condenser and vented tumble dryers was compared, finding that real loads release surprisingly high levels of microfibers (total 341.5 ± 126.0 ppm for those dried in a condenser dryer and 256.0 ± 74.2 ppm for those dried in a vented dryer), similar in quantity to microfibers produced during the first highly-shedding drying cycle of a new T-shirt load (total 321.4 ± 11.2 ppm) in a condenser dryer. Vented dryers were found to be significant contributors to waterborne microfiber pollution if consumers clean the lint filter with water in accordance with some published appliance usage instructions, as most (86.1 ± 5.5% for the real consumer loads tested) of the microfibers generated during vented tumble drying were collected on the lint filter. Therefore, tumble dryers are a significant source of waterborne and (for vented dryers) airborne microfiber pollution. While reducing the pore size of tumble dryer lint filters and instructing consumers to dispose of fibers collected on lint filters as municipal solid waste could help reduce the issue, more sophisticated engineering solutions will likely be required to achieve a more comprehensive solution.

## Introduction

### The impact of fabric care on microfiber pollution

Humans have a close relationship with textiles that begins shortly after birth and continues to death. Apparel textiles surround our body, and non-apparel textiles such as towels, bedding,

of Launderers provided a grant to AMC through the Master (2022) of that institution and its Education Committee. No grant number was provided. The Worshipful Company of Launderers had no role in study design, data collection and analysis, decision to publish, or preparation of the manuscript. https://www.launderers.co.uk/ Northumbria University funded the study through employment of JRD and SL, and provision of consumables. Only the co-authors affiliated to this institution were involved in study design, data collection and analysis, decision to publish, and preparation of the manuscript. https://www.northumbria.ac.uk/ Procter & Gamble Technical Centres Ltd provided funding in the form of salaries for NJL, AKM and AJS and purchase of appliances and related laboratory consumables. In addition to NJL, AKM and AJS, another member of Procter & Gamble staff contributed to the study as described in the acknowledgements but only these individuals were involved in study design, data collection and analysis, decision to publish, and preparation of the manuscript. Procter & Gamble management gave approval to publish, but this process did not influence the text of the manuscript. https://www.pg.com/.

**Competing interests:** I have read the journal's policy and the authors of this manuscript have the following competing interests: NJL, AKM and AJS are employed by Procter & Gamble Technical Centres Ltd, a wholly owned subsidiary of the Procter & Gamble Company. Procter & Gamble is a manufacturer of fabric care products such as laundry detergents, fabric conditioners and dryer sheets. This does not alter our adherence to all PLOS ONE policies on sharing data and materials.

curtains, and upholstery bring important functional and esthetic benefits to our lives. This relationship is fueled by a vast textile industry which now exceeds global annual per-capita production of 13 kg with annual apparel production projected to reach 102 million tonnes by 2030 [1]. While early textiles comprised simple natural fibers, perhaps colored with natural dyes, technological advancement since the first industrial revolution has led to a complex global textile landscape with increasingly sophisticated fabrics. This has included developments in fiber technology, especially the introduction of synthetic fibers such as polyester, polyamide, elastane (spandex) and acrylic, as well as wood-based regenerated cellulose fibers such as viscose and lyocell and the continued use of cotton, wool, silk, and other natural fibers. Advances have also been made in textile spinning, weaving, and knitting, but particularly in dyeing and finishing processes which can fundamentally change the properties of the fabric versus those expected from the constituent fibers.

In recent years, concerns have been raised about the environmental impact of the textile industry in terms of natural resource utilization, $CO_2$ emissions, water usage, water and air pollution, and disposal of textile waste. Fibers are the most basic component of textiles, and there are increasing concerns about the environmental and human safety of fibers that are lost during the wear and care of textiles. Browne et al. [2] first reported concerns that the washing of textiles liberates microfibers that can ultimately pollute aquatic ecosystems. While initially the focus was on non-biodegradable synthetic fibers, as natural fibers such as cotton are considered to be biodegradable [3], studies showing that dyes and finishes slow down the rate of natural fiber biodegradation [4] suggest a need to reduce all forms of aquatic microfiber pollution rather than synthetic microfibers only.

Many groups have studied the drivers of waterborne microfiber release during laundering including the relevance of textile construction, washing conditions and fabric care products, and this area has been comprehensively reviewed [5–9]. Some washing machines are now being fitted with microfiber filters, others have washing cycles designed to minimize microfiber release, and various filtration devices have been developed that can be fitted to existing washing machines with mixed results [10]. Very recently, more sophisticated biomimetic filtration systems have been developed [11] that are designed to have low manufacturing costs, high effectiveness, and ease of use. Now that France has introduced legislation requiring all new washing machines sold from 2025 [12] are installed with a microfiber filter and similar legislation is being considered in the U.K. and elsewhere, it is hoped that the combination of microfiber removal from washing machines and advances in textile design to reduce shedding will contribute to significant reductions in microfiber emissions from washing machines in coming decades.

However, microfibers are also understood to have environmental impacts beyond aquatic ecosystems, with terrestrial and air pollution emerging as areas of concern [13–16]. Reports of microplastics being found in human lung tissue [17] suggests that the mechanisms of airborne fiber loss through wear and care of fabrics requires further investigation. It has long been accepted as a principle of forensic science that clothing transfers fibers through surface contact, but it is now accepted that this transfer can occur through air without any surface contact [18] raising concerns that significant airborne microfiber pollution could be occurring during everyday wearing [19] and especially during textile drying whether on lines or through use of a tumble dryer.

## Contribution of tumble drying

O'Brien et al. [20] reported that tumble drying causes airborne microfiber pollution through air sampling tests during use of an indoor-vented tumble dryer. This was confirmed by Kapp

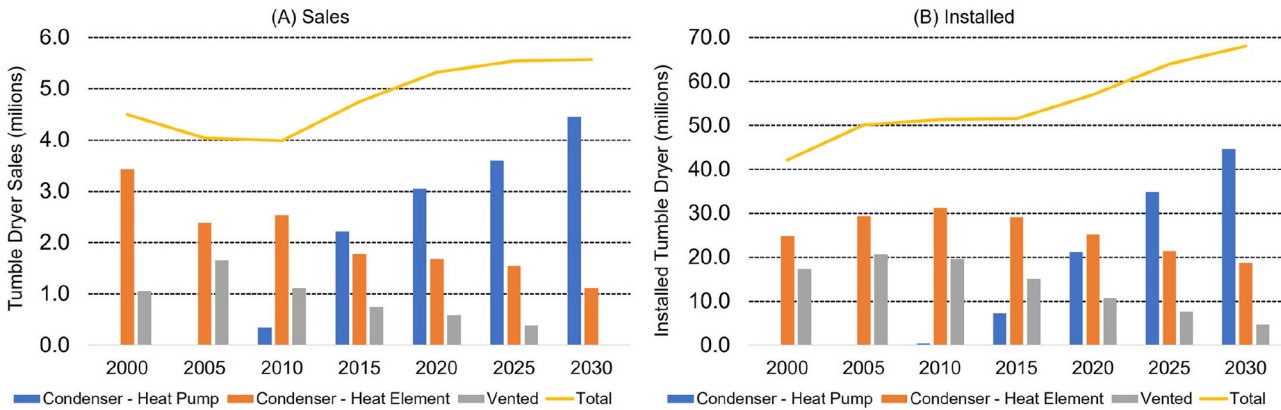

**Fig 1. Segmentation of European tumble dryer sales and installed appliances.** Numbers of appliances sold (A) and installed (B) in Europe are given in millions, split between condenser heat-pump, condenser heat-element and vented.

and Miller [21] using fiber deposition onto snow in the vicinity of the exhaust vent as proof of airborne transfer and subsequent terrestrial pollution. Further studies describing microfiber release from tumble dryers were reported by Kärkkäinen and Sillanpää [22] but these were focused on fiber collection by the lint filter, it was not clear whether their work involved a vented or condenser dryer, and waterborne- or airborne-release was not considered. Further work by Tao et al. [23] quantified airborne emission of microfibers from vented dryers reporting 433,128–561,810 microfibers to air during 15 minutes of use, although this work involved a commercial tumble dryer (Electrolux Wascator TT200). Lant et al. [24] conducted further studies using a domestic vented tumble dryer, also considering the relevance of the built-in lint filter designed to remove fibers from the exhaust air and impact of fabric care products. This found that microfiber emissions from the vented dryer tested were significant and involved similar quantities to those released to the drain during the preceding wash cycle. Results also showed that redesign of the lint filter could significantly reduce the quantities of microfibers released through the exhaust, and that use of liquid fabric softener and/or dryer sheets can also reduce emissions by either improving the effectiveness of the lint filtration process (liquid fabric softener) or acting as a magnet for microfibers in the drum of the dryer (dryer sheets). This article suggested that a move away from vented tumble dryers to fully sealed dryer designs that have no exhaust outlet could be a useful step in preventing such airborne microfiber pollution. While vented dryers remain very common in North America, in Europe the market is steadily moving away from these appliances towards non-venting condenser appliances as shown in Fig 1, based on actual and projected data reported by Maya-Drysdale et al. [25].

While there is little doubt that non-venting dryers will cause less airborne microfiber pollution than vented dryers, there is a risk that these condenser and heat-pump dryers might involve other sources of microfiber pollution. Like vented dryers, condenser dryers have a lint filter designed to remove microfibers from the air flow, but any fibers that aren't removed in this way will be collected on the condenser or in the condensed water which is either drained away or collected within the machine for disposal or reuse. This choice of whether to directly connect the condenser water outlet to a drain or keep it internally connected to a water collection compartment in the dryer is made by the consumer depending on their own preference and proximity to drain pipework. In addition, manufacturer instructions recommend regular cleaning of the condenser, typically by rinsing it with flowing water in a sink. Some dryers

contain a 'self-cleaning condenser' which avoids this need to clean the condenser by using the condensate to rinse the condenser. Thus, there is a risk that condenser dryers simply shift the airborne microfiber pollution issue associated with vented dryers towards increased waterborne microfiber pollution through disposal of contaminated condensate or regular cleaning of the condenser in a sink. As both vented and condenser dryers have lint filters, these are also a potential contributor to waterborne microfiber pollution if they are cleaned in water which is one cleaning option suggested by dryer manufacturers.

The initial objective of the present study is to determine the contribution of condenser tumble dryers to waterborne microfiber pollution for the first time, using test loads that comprise a mixture of cotton and polyester garments to reflect the two most common fibers used in machine-washable clothing. This will allow the proportion of these fibers in the load being dried to be compared with fibers collected in the lint filter (typically disposed in municipal solid waste or under running water), on the condenser (typically rinsed under running water and disposed to the drain) and in the condensed water (typically disposed to the drain or recycled within the household, for example to water houseplants).

The second objective is to measure total microfiber release during vented and condenser tumble drying of real laundry loads sourced from U.K. households. This is believed to be the first study involving microfiber release during tumble drying from real consumer wash loads. As real laundry loads tend to contain a mixture of fabric types, of different ages, and may contain residual soils, this will give useful insights into the size of the contribution of tumble drying to airborne and waterborne microfiber pollution under real world conditions.

Previous studies [24] involving vented dryers quantified microfiber collection onto lint filters but assumed that these would be disposed by consumers as municipal solid waste. A further objective of the present study is to survey usage instructions provided by various tumble dryer manufacturers to determine the extent to which consumers are being directed to dispose lint filter microfibers in this way versus alternatives such as cleaning under running water.

The results from these studies will provide useful insights to the appliance industry and legislators seeking to reduce the quantities of microfibers entering the environment from laundry processes, especially the contribution from tumble drying which has not yet been extensively studied nor attracted significant interest from relevant industry associations or legislators.

## Materials and methods

### Textiles and fabric care products

One Ariel 3in1 Pod was used as the detergent in every wash cycle. This product was manufactured by Procter & Gamble and purchased in the U.K. during summer 2022.

Testing with new clean garments involved loads comprising 10 100% cotton T-shirts (Fruit of the Loom Men's Original T-shirt, Product Code 61082, Orange color, size Large) and 12 100% polyester T-shirts (Fruit of the Loom Women's Performance T-shirt, Product Code 61392, Black color, size Extra Large). All T-shirts were supplied by BTC Activewear, Wednesbury, U.K. The loads were weighed prior to washing to enable calculation of microfiber release as a proportion of load mass, and the composition of the loads was 50.4 ± 0.1% cotton and 49.6 ± 0.1% polyester by mass.

Testing with soiled consumer loads involved laundry bundles sourced from volunteers in Newcastle upon Tyne, U.K. in the summer of 2022. Ethical approval for carrying out this research was granted by Northumbria University's Ethics Committee (code RE-CFS-064). Written informed consent forms outlining the details of the research and handling of data were signed by all volunteers. These contained a mixture of washable apparel and household textiles, pre-sorted by the consumers into loads as they would for laundering at home. Each

load was weighed prior to washing to enable calculation of microfiber release as a proportion of load mass. The washed and dried loads were returned to the consumers immediately after testing. Photographs of the soiled laundry loads used are given in S1 Fig.

## Washing and drying procedure

**Washing protocol.**   All wash testing involved European washing conditions conducted using 6.7 grains per U.S. gallon hardness water and a set of four Miele W3922 washing machines using the 30˚C Cotton Short program (90 minutes total duration, 1600 rpm spin speed). The washing machines were thoroughly cleaned between tests using washout cycles as described by Lant et al. [24, 26]. Testing with new clean wash loads involved four consecutive washing and drying cycles and three replicates, i.e., each of three separate wash loads comprising 22 T-shirts (10 cotton, 12 polyester) was washed and dried four times. Testing with soiled consumer loads involved a single wash and dry cycle, completed with a total of 16 loads, eight dried with a condenser dryer and eight dried with a vented dryer.

**Drying protocol.**   Condenser tumble drying was conducted after each washing cycle using an Indesit condenser tumble dryer (model I2D81WUK, supplied by AO Retail Ltd, Bolton, U. K.) for one hour on the high heat setting. Mechanical and thermal energy settings were the same for each load. The condenser tumble dryer was cleaned of residual fibers prior to testing, and between loads, by drying a set of nine sponges (Super Bright brand, item code SU37, supplied by Amazon U.K.) soaked in 1 liter of water double bagged within polyester drawstring mesh laundry bags (OTraki brand, 20 x 24 inch, supplied by Amazon U.K.). The condenser dryer was then run for a further 30 mins on high heat, before thorough cleaning of the lint filter and its housing (using a Nilfisk high efficiency vacuum cleaner), cleaning of the condenser (using copious amounts of tap water) and placement of a clean container under the drain hose to collect condensed water.

Vented tumble drying was conducted after each cycle using an Indesit vented tumble dryer (model IDV75, supplied by AO Retail Limited, Bolton, U.K.) for one hour on the high heat setting. The vented tumble dryer was cleaned prior to testing, and between loads, by running a cycle (60 mins) without any garments inside followed by thorough cleaning of the lint filter and its housing (using a Nilfisk high efficiency vacuum cleaner).

The tumble dryers were placed on a balance to measure the drying process in real time, located in a specialized laboratory for appliance research with high levels of ventilation. Internal temperature (using a probe) and power consumption (at the power supply outlet) were recorded during drying to ensure good consistency between loads. The measurement of dryer mass during the drying process confirmed that loads were completely dry after 1 hour.

## Microfiber collection during the drying cycle

**Condenser dryer.**   Evaluation of the microfiber release from the condenser dryer (Fig 2) was carried out at three locations:

1. The **lint filter** which was washed in copious amounts of clean water to recover all the fibers.

2. The **condenser** where accumulated fibers were recovered by washing in copious amounts of clean water.

3. The condensed **water**.

**Vented dryer.**   Evaluation of the microfiber release from the vented dryer (Fig 3) was carried out at two locations:

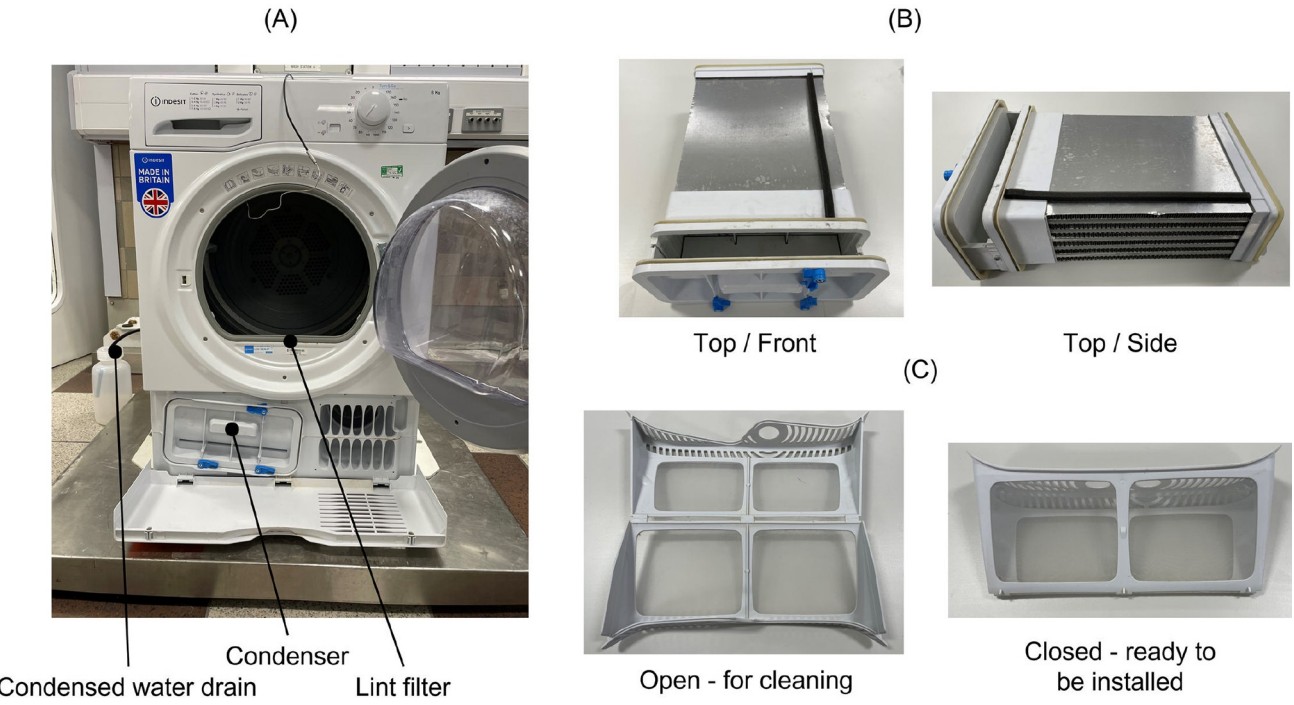

**Fig 2. Condenser tumble dryer.** Condenser dryer (**A**) showing the collection of water from the drain hose and location of the lint filter and condenser housings. The condenser (**B**) and lint filter (**C**) are also shown separately.

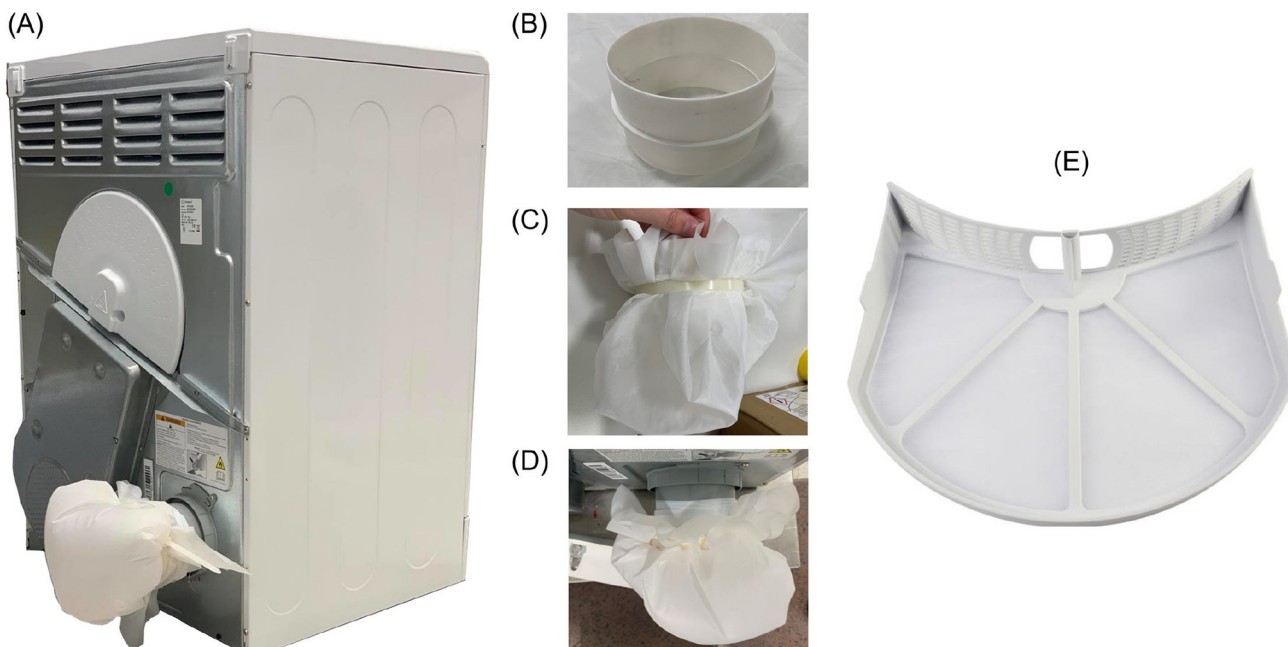

**Fig 3. Vented tumble dryer.** Rear of tumble dryer (**A**) with 20 μm CellMicroSieve installed to collect microfibers from the exhaust. The CellMicroSieve is secured to a 100 mm plastic pipe connector (**B**) using a cable tie as shown in (**C**). The pipe connecter was then secured to the dryer exhaust pipe using electrical tape as shown in (**D**). The lint filter is shown separately in (**E**).

1. The **lint filter** which was washed in copious amounts of clean water to recover all the fibers.

2. Fibers released through the exhaust **vent**.

The fibers released from the exhaust vent were collected as described in Lant et al. [24] using the setup shown in Fig 3. This involved microfibers being collected using a 20 μm Cell-MicroSieve (BioDesign Inc., Carmel, N.Y., U.S.A.), attached to the dryer exhaust using a 100 mm plastic pipe connector (Fig 3A) (model 414c, Manrose Manufacturing Ltd., U.K.). The CellMicroSieve was connected to one side of the plastic pipe connector (Fig 3B) using 450 mm long, 10 mm wide cable ties (product 90526, Screwfix Direct Ltd., U.K.) as shown in Fig 3C and then connected to the vent pipe using electrical tape as shown in Fig 3D. At the end of the drying, the CellMicroSieve was thoroughly washed with clean water to re-suspend the collected fibers.

## Microfiber analysis

**Mass measurement.**   Microfiber suspensions were filtered onto pre-weighed Whatman No 541 filter paper (G.E. Life Sciences, Little Chalfont, U.K.) using a Büchner funnel under vacuum before drying overnight at 40°C.

The mass of collected fibers was then calculated, corrected for the percentage loss in filter paper weight on drying, which was determined by recording the mean percentage mass loss on drying of 10 similar papers. Microfiber release data are presented in parts per million (ppm), i.e., as mg of released fiber per kg initial dry fabric load mass.

**Fiber composition and size.**   Fibers were randomly recovered from each filter paper, using a mask of diameter 2 cm (S2 Fig). Recovery was achieved by pressing the high-adhesive tape (Forensic Alliance Tape, 190 x 70 mm, 3L Office Products A/S, Tommerup, Denmark) on the 2 cm diameter sampling point. Initially, a single thumb press on the external surface of the tape allowed a localized pressure to be exerted; this was followed by additional pressure by dragging over the same sampling point a 500 g weight. The dragging of the 500 g weight allowed a constant pressure to be applied to aid fiber recovery on the tape. The tape was then carefully lifted and attached to a transparent A5 acetate sheet, which had been previously appropriately labelled. The entire recovery process with tape was repeated between 1 and 24 times (S2 Fig), based on the exhaustive recovery of fibers, as evidenced by negligible fiber recovery on the condensate filter papers through to significant fiber recovery on the lint filter filtrate water. Then, a fiber lifted tape containing a medium-loaded number of fibers was selected for further analysis and identification. The medium-loaded fiber acetate sheet was then further sub-sampled using a coning and quartering principle to reduce the number of fibers to less than a maximum of 200. The typical numbers of sub-sampled fibers for the condenser dryer from the T-shirts varied; for the lint filter between 29 and 189, for the condenser between 7 and 192, and for the condensed water between 2 and 159.

Individual fibers were recovered using stainless steel HP Grade tweezers, type AGT1525 and AGT5506 (Agar Scientific Ltd., Stansted, U.K.). All the recovered fibers were removed and mounted individually on labelled glass microscope slides (CIMED, 1–1.2 mm thick, 25 x 75 mm) using Aquatex (Merck, Sigma-Aldrich, Poole, UK) and covered with round cover slips (9 mm, Thermo Scientific, Germany); each microscope slide contained up to 10 fibers. Recovery of fibers was carried out manually under a Low Power Microscope (x 25 magnification, Leica S6E, Leica, Milton Keynes, U.K.). High Power Microscopy (x 400 magnification, Olympus CX23, Olympus, Hamburg, Germany) was used to confirm identify of cotton (orange fibers) and polyester (black fibers). Use of an Olympus CX23 microscope coupled

with a Euromex camera with Image Focus 4.0 software allowed measurements of fiber length (in mm) and width in μm).

**Statistics.** Where mean data is accompanied by a range, e.g., 4.1 ± 1.2, the latter number is the standard deviation. Statistical significance was determined using Student's t-test; comparisons with a p-value of <0.050 were considered to be significantly different at 95% confidence level. All statistical analysis was conducted using Microsoft Excel.

## Results and discussion

### Microfiber release from clean T-shirts in condenser tumble dryers

**Gravimetric quantification.** The loads of 10 cotton and 12 polyester new and clean T-shirts were subjected to four consecutive washing and drying cycles with collection of fibers from the lint filter, condenser and condensed water during each drying cycle as described in the previous section. The loads each had a mass of 2.84 ± 0.01 kg. Results, summarized in Fig 4 and given in full in S1 Table show that total microfiber release (i.e., the sum of fibers collected on the lint filter, condenser and in the condensed water) significantly reduces with each cycle. This is in line with previous reports that new garments show relatively high levels of shedding in both the washing and tumble drying stages [24, 26] with levels reducing during the following cycles. The lint filter, designed to stop fibers from getting onto the condenser or into the condensed water shows mean effectiveness of 91.8 ± 1.4% (n = 12), i.e., over eight percent of fibers are passing through the filter onto the condenser or into the condensed water. The lint filter shows similar efficiency (defined as the quantity of microfibers collected as a percentage

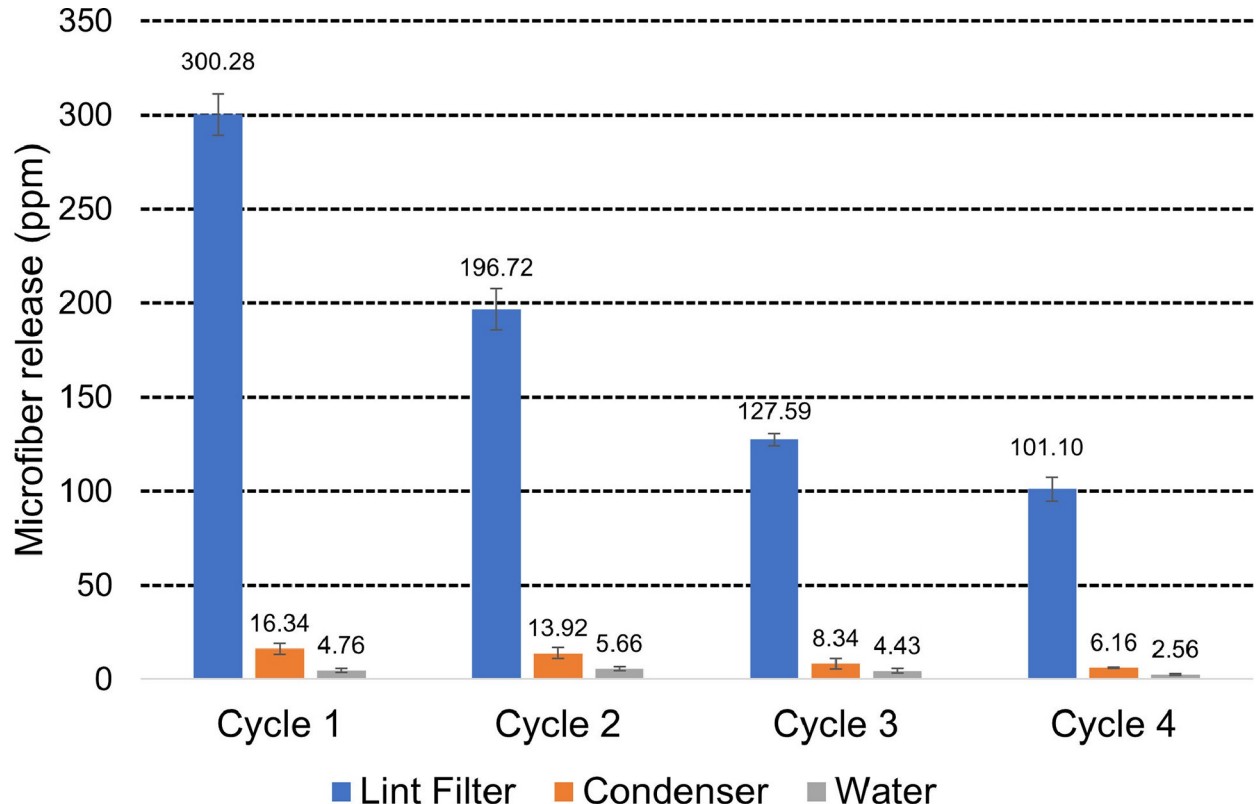

**Fig 4. Microfiber release from clean loads in condenser tumble dryers.** Microfiber release at three measurement points for each of four cycles is given in ppm (mg release per kg fabric). Error bars are the standard deviation.

of total microfibers released in the appliance) in the first cycle (93.4 ± 1.1%, n = 3), second (91.0 ± 1.4%, n = 3), third (90.9 ± 1.2%, n = 3) and fourth (92.0 ± 0.6%, n = 3) cycles.

The quantity of microfibers trapped on the lint filter significantly reduces with each wash cycle and similar trends are found on the condenser and in the condensed water (S1 Table). Mean levels of microfiber accumulation on the condenser and in the condensed water over the four cycles was 11.2 ± 4.7 ppm (n = 12) and 4.4 ± 1.6 ppm (n = 12), respectively, suggesting that washing the condenser is likely to be a bigger contributor to aquatic pollution than disposal of the condensed water. However, both sources need to be considered, especially because consumers are more likely to recycle the condensed water for other purposes in the home.

The manufacturer's instructions for the condenser dryer used in our trials (Indesit I2D81WUK) recommends that the lint filter be cleaned "under running water or using a vacuum cleaner". If consumers follow the former instruction and clean the lint filter in a sink, our data suggests that this would have a significant impact on waterborne pollution arising from tumble drying because the mean level of microfibers collected on the lint filter over four cycles was 181.4 ± 80.9 ppm (n = 12), far greater than the above levels of 11.2 ± 4.7 ppm (n = 12) and 4.4 ± 1.6 ppm (n = 12) collected on the condenser and in the condensed water, respectively. In the conditions of our test, a consumer who disposes of all three fiber sources in the drain (sum of means 197.0 ppm per wash) would release 12.6 times more fibers than the 15.6 ppm from a consumer who discarded fibers from the lint filter in municipal solid waste and only rinsed the fibers on the condenser and in the condensed water down the drain.

**Fiber composition and dimensions.** The released fibers collected at the lint filter, on the condenser and in the condensed water were analyzed to determine the ratio of cotton to polyester fibers compared to the 50.4% cotton and 49.6% polyester (by weight) of the loads used. Results summarized in Fig 5 and given in full in S2 Table show that the composition (by number) of fibers shows some differences between point of collection. On average (n = 12) over the four cycles, the fibers are 64.7 ± 6.8% cotton on the lint filter, 69.9 ± 9.9% cotton on the condenser and 82.9 ± 8.2% cotton in the condenser water. While the increase in cotton level between lint filter and condenser stages is not statistically significant, the high proportion of cotton fibers observed in the condenser water is highly significant versus lint filter and condenser stages (t-test p = 0.00 for both). This observation that the lint filter is better at removing polyester fibers than cotton fibers (comparison of fiber content on the filter and at collection points after it) is in line with results reported previously in the context of vented dryers [24].

The dimensions of fibers sampled at each of the three measurement points is summarized in Fig 6 and given in full in S3 Table. The cotton fibers analyzed (total n = 240) were 19.6 ± 3.8 μm wide, significantly (t-test p = 0.00) thicker than the polyester fibers (total n = 240) at 13.7 ± 2.0 μm. However, there were no significant differences in the widths of each fiber type between the three different measurement points in the appliance or between the four cycles.

The cotton fibers analyzed (total n = 240) were 0.83 ± 0.47 mm long, slightly shorter than the length of polyester fibers (total n = 240) at 0.90 ± 0.35 mm. The slightly longer length of the polyester fibers might help explain the relatively high level of efficiency of the lint filter in collecting this fiber relative to cotton, as discussed previously. It is evident from these data that the microfibers detected comprise broken fiber fragments rather than loss of complete fibers from the garments given that even short staple cotton has a fiber length of greater than 0.9 cm, and the polyester filaments in the performance T-shirts are likely to be even longer. This discrepancy in length between the textile and released microfiber fragments suggests that the high levels of microfiber release in the first few cycles of washing and drying is likely to be driven by release of small microfiber debris from within the yarns that were present in the new textile. Previous extended wash tests have found that microfiber release reaches a low plateau

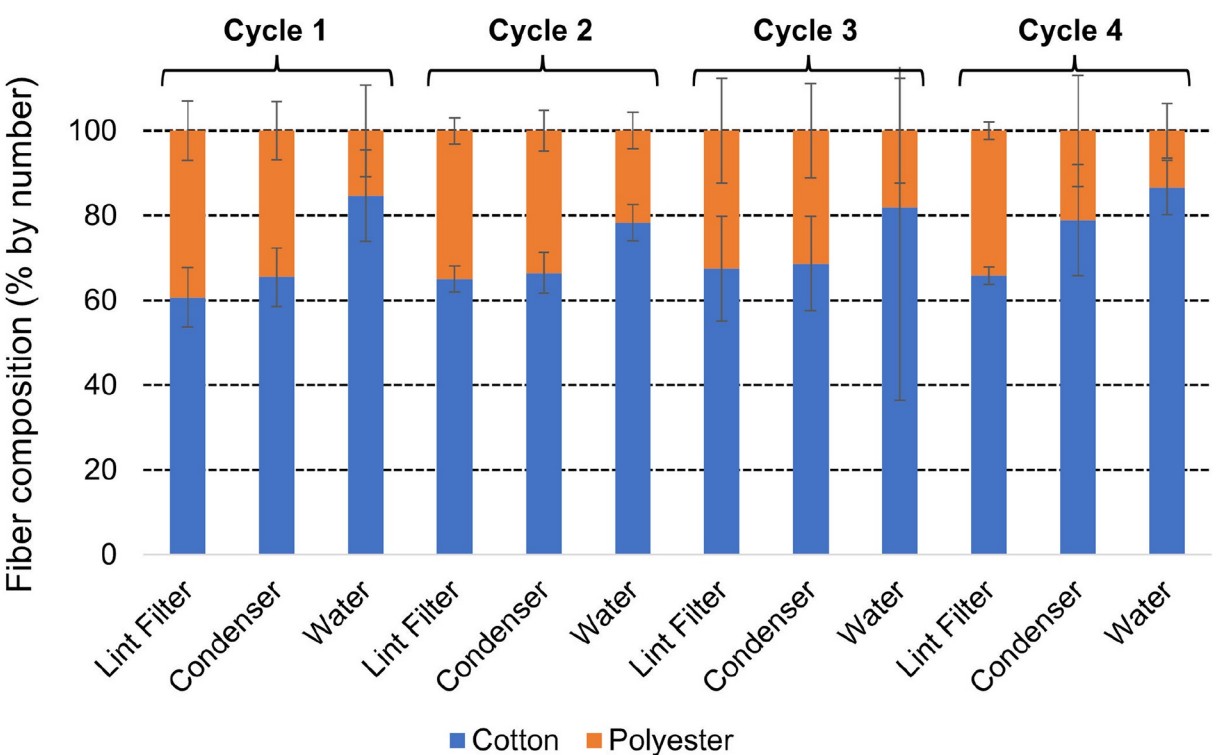

**Fig 5. Fiber composition of microfibers release from clean loads in condenser tumble dryers.** Levels of cotton and polyester fibers at three measurement points over four cycles is given in percentages (by number). Error bars are the standard deviation.

after around eight wash cycles [26], suggesting that this may be the point at which the initial microfiber debris in the new textile has been essentially removed and further microfibers released from that point result from new fiber breakage and unplucking processes.

Analysis of the fiber lengths at the three different collection points showed some significant differences. In line with expectation, cotton fibers collected on the lint filter across all four cycles (n = 80) were 0.97 ± 0.49 mm long and significantly (t-test p = 0.05) longer than those

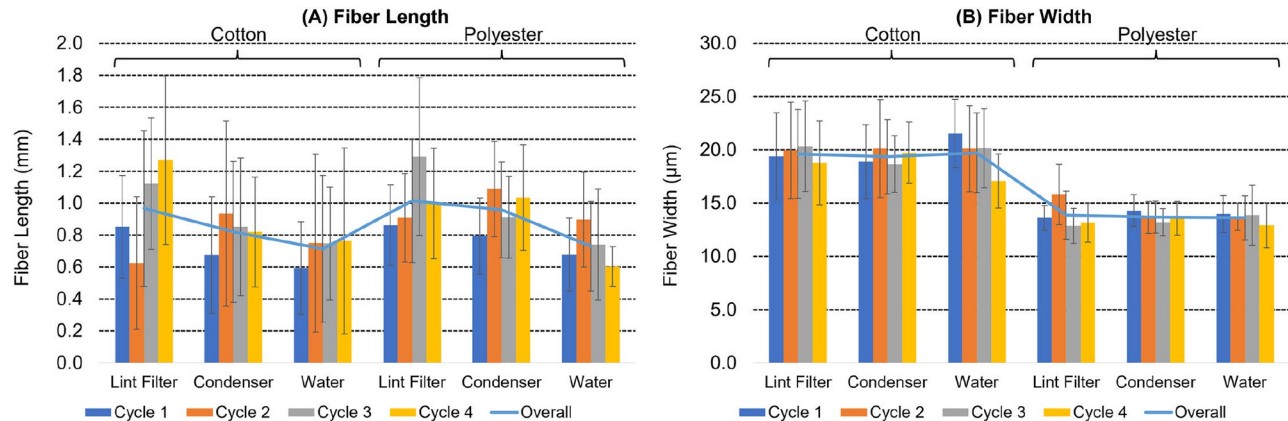

**Fig 6. Fiber dimensions of microfibers released from clean loads in condenser tumble dryers.** Fiber lengths (A, in mm) and widths (B, in μm) are given for the cotton and polyester fibers collected at three measurement points over four cycles. Error bars are the standard deviation.

collected on either the condenser (n = 80, 0.82 ± 0.44 mm) or (t-test p = 0.00) in the condensed water (n = 80, 0.71 ± 0.46 mm). Cotton fibers collected on the condenser were not significantly different in length from those collected in the condensed water.

Polyester fibers collected on the lint filter across all four cycles (n = 80) were 1.01 ± 0.39 mm long a similar length to those collected on the condenser (n = 80, 0.96 ± 0.30 mm). However, fibers collected in the condensed water (n = 80, 0.73 ± 0.28 mm) were significantly shorter than those collected on the lint filter (t-test p = 0.00) or condenser (t-test p = 0.00).

No significant differences in length of released microfibers were observed across the four cycles tested at each point of collection.

## Microfiber release from real consumer wash loads in both condenser and vented tumble dryers

**Gravimetric quantification.** Most of the research into the release of microfibers during laundry processes involves wash loads that are clean and comprise fabrics of the same age. In reality, consumers typically wash a variety of items of different ages together that are soiled with incidental stains (e.g., grass, tea, wine, vegetable oil) and body soiling (e.g., sebum, sweat, fecal, urine, menstrual fluid) and contain a mixture of textile fibers and finishes. These soils are rarely completely removed during the laundering process, especially if poor quality detergents are used. Building on previous studies of waterborne microfiber pollution arising from washing real wash loads [26], the present study sought to shift focus to the impact of tumble drying similar loads, involving eight loads dried in a vented dryer (with measures of microfiber collection on the lint filter and airborne release from the exhaust vent) and eight loads dried in a condenser dryer (with measures of microfiber collection on the lint filter, collection on condenser and present in the condensed water).

The soiled laundry test loads sourced from U.K. consumers were washed once and dried with collection and gravimetric measurement of fibers as described previously. Mean masses of the wash loads used was 2.74 ± 0.72 kg (n = 8) for those dried in a condenser dryer and 2.75 ± 1.05 kg (n = 8) for those dried in a vented dryer. Microfiber release results are summarized in Fig 7 and given in full in S4 Table. The results show that total release of microfibers from the loads was 341.5 ± 126.0 ppm for those dried in the condenser dryer (sum of 313.3 ± 120.8 ppm collected on the lint filter, 21.6 ± 7.1 ppm on the condenser and 6.7 ± 2.0) and 256.0 ± 74.2 ppm for those dried in the vented dryer (sum of 222.7 ± 74.3 ppm collected on the lint filter and 33.3 ± 13.0 ppm released through the vent). Given that the same condenser dryer was used to dry these loads and the new T-shirt loads described previously, we can directly compare release levels between these fabric loads. The soiled consumer loads appear to show microfiber release levels in the condenser dryer that are similar to the first, highly shedding, drying cycle of the T-shirt load (321.4 ± 11.2 ppm, the sum of 300.3 ± 11.1 ppm collected on the lint filter, 16.3 ± 2.8 ppm collected on the condenser and 4.8 ± 1.2 ppm collected in the condensed water).

This finding that soiled mixed consumer loads release relatively high levels of microfibers during tumble drying is consistent with the previous report [26] describing waterborne microfiber pollution levels arising from the washing of similar consumer loads sourced from consumers in the U.K. They reported an average of 114 ± 66.8 ppm of waterborne microfiber release from testing involving 79 real wash loads. Integration of those data regarding waterborne release from washing machines, and the present study of release from dryers suggests that the total contribution of washing and drying in a condenser dryer on waterborne pollution will heavily depend on consumer behavior regarding the cleaning of the lint filter. This is illustrated by Fig 8 which combines the previous washing machine release data [26] with the

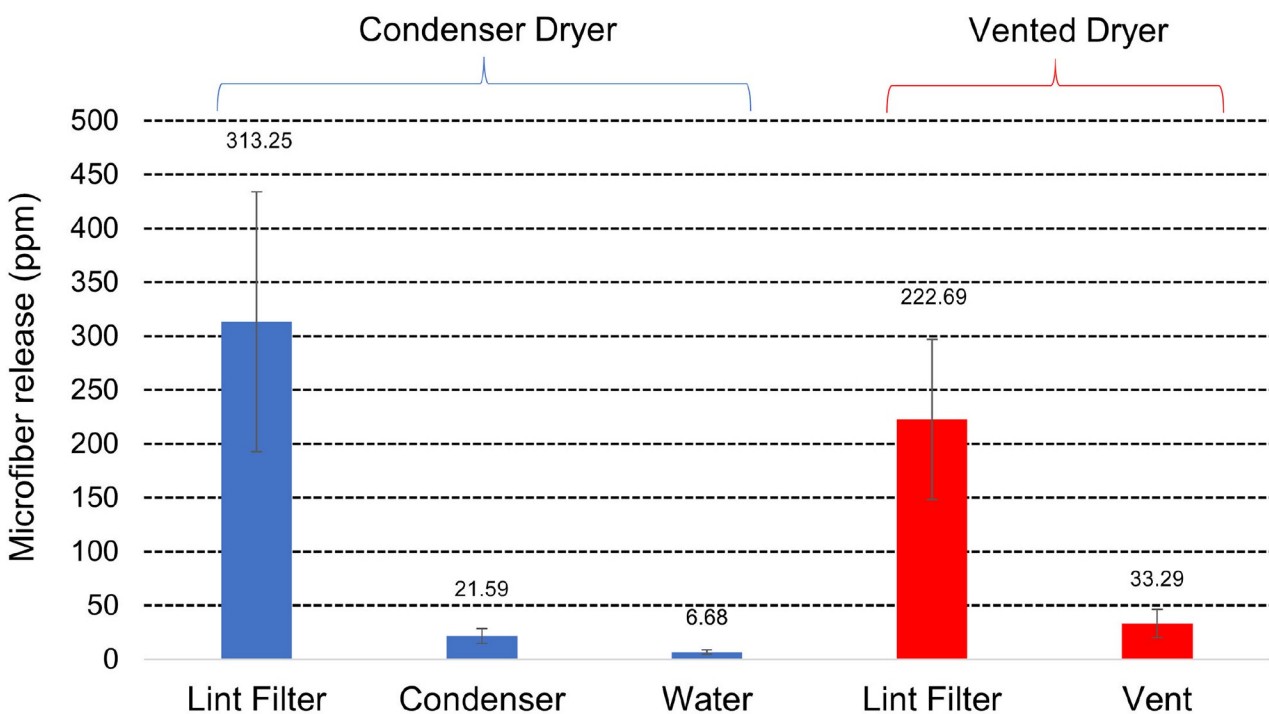

**Fig 7. Microfiber release from soiled consumer loads.** Microfiber release at three (condenser dryer) or two (vented dryer) measurement points is given in ppm (mg release per kg fabric). Error bars are the standard deviation.

present study to estimate the total microfiber release levels to water and air from washing and drying clothes as a function of dryer type and method of lint filter fiber disposal. This shows that among consumers that dispose of tumble dryer lint filter fibers to the drain (in full compliance with the usage instructions of both dryers tested), this will be a bigger contributor to waterborne microfiber pollution than the combination of emissions from the preceding washing cycle, rinsing the tumble dryer condenser and disposing of the condensed water down the drain. These consumers therefore have the potential to reduce their contribution to microfiber pollution by over half by changing behavior and disposing of lint filter fibers in municipal solid waste. While the data in Fig 8 provide useful guidance, we need to keep in mind that they were generated using a single model of washing machine, condenser dryer and tumble dryer and single operating program for each. Still, they provide the first evidence that condenser tumble dryers make a significant contribution to waterborne microfiber pollution and that vented dryers can contribute to both airborne and waterborne pollution depending on how the consumer cleans the lint filter.

**Efficiency of lint filters.** The lint filter of the condenser dryer tested showed efficiency of 91.1 ± 3.4%, significantly (t-test p = 0.05) higher than that of the vented dryer tested at 86.1 ± 5.5% (see S4 Table). Light microscopy of the lint filters (Leica MZ16A with PlanApo 1.0x objective lens, Leica Microsystems, Milton Keynes, U.K.) from the condenser and vented dryers are shown in Fig 9, showing that the pore size of the condenser filter is 0.048 ± 0.002 mm$^2$ compared to the vented dryer at 0.20 ± 0.01 mm$^2$. Given the previous findings [24] that decreasing the pore size of lint filters in tumble dryers significantly improves microfiber filtration, it is likely that the smaller pore size of the condenser dryer explains its superior lint filter efficiency versus the vented dryer.

**Manufacturers' instructions.** While the manufacturer of the condenser and vented dryers used in the present study recommend that fibers collected on the lint filter are removed

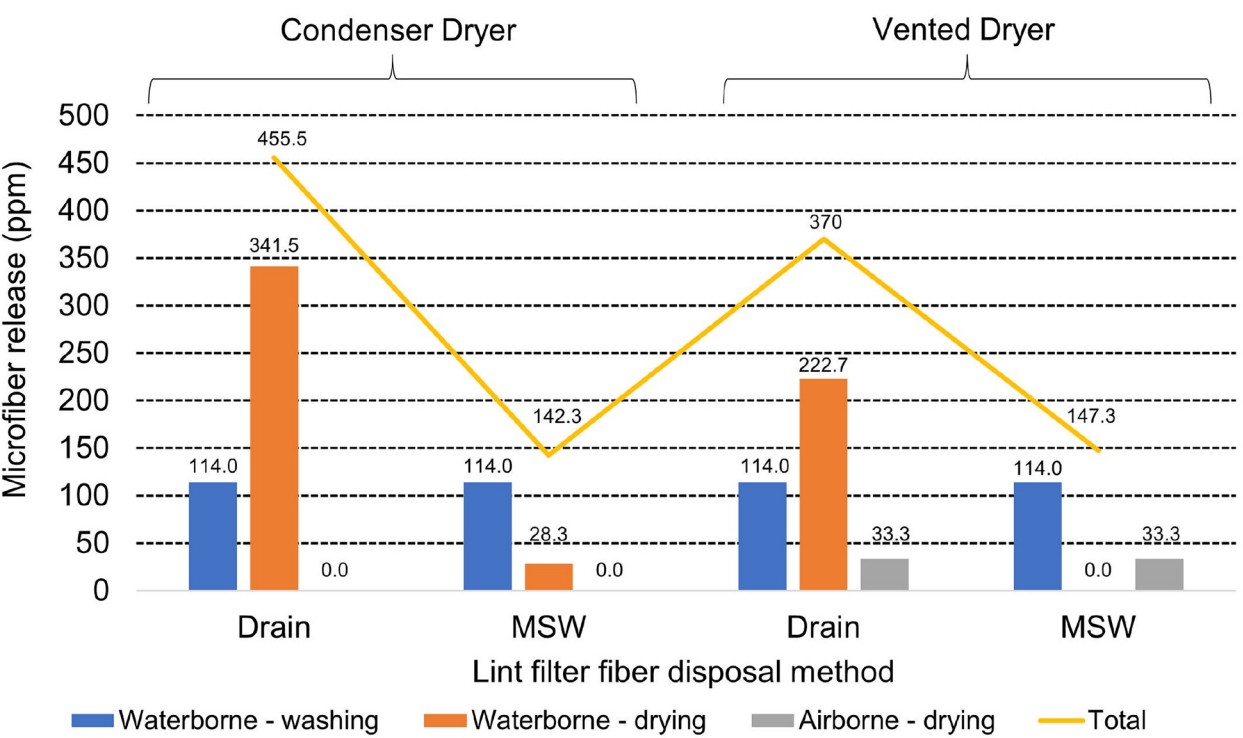

**Fig 8. Estimated microfiber release levels to water from washing clothes and drying in a condenser or vented tumble dryer.** Microfiber release to water (from both washing and drying processes) and air (from drying with a vented dryer) is given in ppm (mg release per kg fabric), comparing use of a condenser dryer with a vented dryer and the impact of lint filter fiber disposal method (MSW = Municipal Solid Waste).

"under running water or using a vacuum cleaner", other dryers are sold with different guidance. S5 Table summarizes a survey of dryers sold in the U.K. in January 2023 with a representative model selected from 24 different brands sold by AO.com (AO Retail Ltd), Currys plc and John Lewis Partnership plc. Although this is only a snapshot of the market, four brands

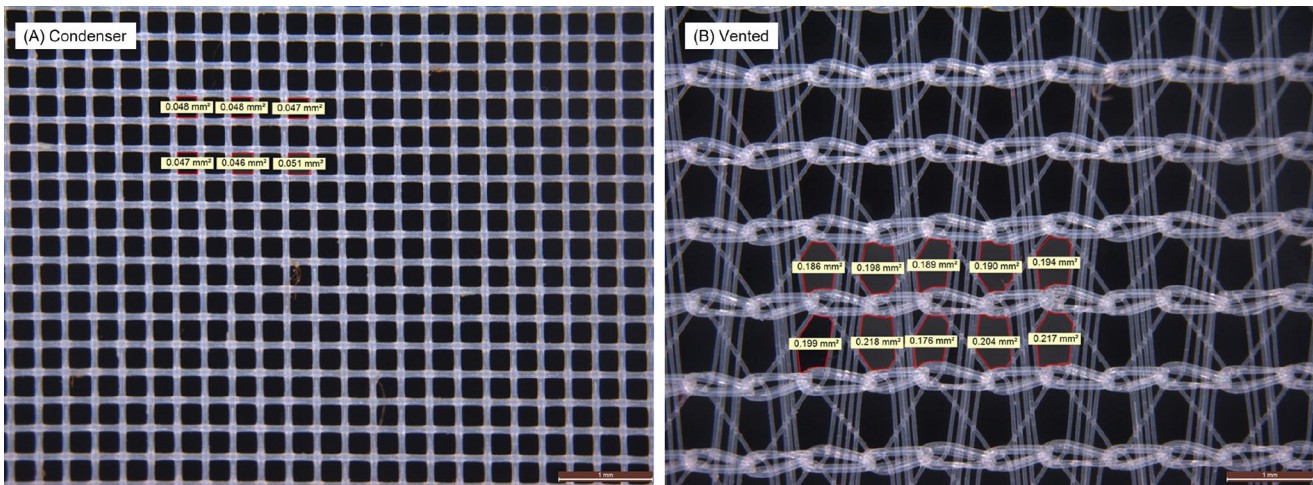

**Fig 9.** Light microscopy of lint filters from the condenser dryer (A) and vented dryer (B). Regions outlined in red are the six (condenser) and 10 (vented) areas used to calculate pore area. A 1 mm scale bar is given on each image.

(Indesit, Hotpoint, Bosch, Neff) appear to recommend use of water for lint removal, with the latter two also advising cleaning using a dishwasher. Two brands (Haier and LG) did not provide any specific guidance whereas all the others specifically recommended methods that do not require water such as removal by hand, vacuum cleaner, cloth or soft brush (provided with the appliance in one case). One brand (John Lewis) explicitly states that water should not be used to clean the lint filter for environmental reasons and that the fibers should be disposed in household waste. Thus, there is significant inconsistency across the dryer market regarding the cleaning of lint filters, creating an opportunity to align on best practices and deliver a more consistent and environmentally-friendly message to consumers.

## Conclusions

Tumble drying is an important part of the laundry process in many countries of the world, as it offers a fast and convenient way of drying clothes. However, tumble dryers can be important sources of both waterborne and airborne microfiber pollution, depending on the type of appliance and consumer behavior regarding disposal of lint filter fibers.

Use of condenser dryers leads to accumulation of microfibers on the lint filter, on the condenser and in the condensed water. Some appliance manufacturers are currently instructing consumers that it is acceptable to rinse the lint filter in a sink. The present study suggests that this advice should be updated to specifically recommend that consumers refrain from cleaning lint filters in this way, in favor of collecting the fibers by hand or using a vacuum cleaner to enable safe disposal in municipal solid waste. However, this will not prevent leakage to the condenser and condensed water, destinations that have a high likelihood of ultimately being discharged to drains. This suggests that the appliance industry might need to rethink the overall problem of fiber management in condenser dryers. A previous study [24] focused on vented dryers found that lint filter efficiency could be improved through reduced pore size, although more sophisticated filtration system designs could be even more effective. Removing fiber contamination from the condensed water also could enable this water to be recycled for a broader range of uses, for example as distilled water for steam irons.

Use of vented dryers is already significantly declining in Europe, a trend that is likely to be accelerated by recent rises in energy prices as more consumers move to heat-pump condenser dryers which are far more energy efficient and kinder to clothes (due to their lower operating temperature), albeit with significantly increased drying times. Although we did not include heat pump dryers in the present study, they share similar waterborne pollution risks to condenser dryers because they also generate condensed water and contain lint filters, condensers and evaporators that collect fibers and need to be cleaned by the consumer.

Vented dryers continue to be sold in Europe and remain very popular in other markets such as North America. The present study suggests that the manufacturers of those appliances should clearly instruct consumers to dispose of lint filter fibers in municipal solid waste and consider ways to significantly enhance the efficiency of the lint filtration processes to minimize airborne pollution.

Dual-purpose combined washer dryers are popular in some markets like the United Kingdom, and many are designed to release all fibers emitted during the drying cycle with the condensed water with no fiber collection, although others do have a lint filter. Other dryers are emerging with self-cleaning condensers by channeling fibers from there into the condensed water.

Thus, the appliance industry, its trade associations and legislators should recognize that all types of tumble dryer can be significant contributors to the problem of environmental microfiber pollution and begin efforts to mitigate this issue through revised usage instructions and improved appliance design. Current plans to introduce microfiber filtration systems into

washing machines are expected to reduce the environmental impact of that stage in the laundering process, suggesting that reapplication of similar approaches to tumble dryers is a logical next step. However, these moves do need to consider the ultimate fate of collected microfibers; is their disposal through municipal solid waste a robust solution or merely a stepping stone to further environmental impact?

## Supporting information

**S1 Fig. Photographs of soiled loads.** Photos C1-C8 and V1-V8 show the soiled consumer loads used for the testing with a condenser dryer (C) and vented dryer (V).
(TIF)

**S2 Fig. Process for microfiber analysis.** Microfibers filtered from each point of collection are recovered using adhesive tape and transferred to acetate sheet for determination of fiber composition and dimensions by microscopy.
(TIF)

**S1 Table. Gravimetric quantification of microfiber release from clean T-shirts in condenser tumble dryers.** The table shows measured mass of the wash load used (kg) and microfibers collected (mg) on the dryer lint filter, on the condenser and in the condensed water. These data are used to calculate quantity of microfibers at these three stages in terms of ppm (parts per million, i.e., mg microfiber released per kg dry wash load) and percentage lint filter efficiency for each of the four drying cycles.
(DOCX)

**S2 Table. Fiber composition of microfibers released from clean T-shirts in condenser tumble dryers.** The table shows the analyzed fiber composition of the microfiber samples collected on the lint filter, condenser and in the condensed water. Columns 'Cotton' and 'Polyester' contain the number of fibers of each type identified within the sample. These are used to calculate the percentage composition.
(DOCX)

**S3 Table. Fiber length and width data for microfibers released from clean T-shirts in condenser tumble dryers.** The table shows measured length (mm) and width (μm) of cotton and polyester fibers sampled from each of the three collection points at each of the four drying cycles.
(DOCX)

**S4 Table. Gravimetric quantification of microfiber release from real consumer loads in condenser and vented tumble dryers.** The table shows measured mass of the wash load used (kg) and microfibers collected (mg) for on the dryer lint filter, on the condenser and in the condensed water for the condenser dryer and on the dryer lint filter and from the exhaust vent for the vented dryer. These data are used to calculate quantity of microfibers at these stages in terms of ppm (parts per million, i.e., mg microfiber released per kg dry wash load) and percentage lint filter efficiency for each of the four drying cycles.
(DOCX)

**S5 Table. Survey of lint filter cleaning instructions in tumble dryers.** The table shows details of representative examples of tumble dryers sold under 24 different brands in the UK market, with details of instructions regarding cleaning of lint filter.
(DOCX)

## Acknowledgments

The authors thank William Caufield of Procter & Gamble Newcastle Innovation Center for the microscopy of tumble dryer lint filters.

## Author Contributions

**Conceptualization:** Suzanne Lonsdale, John R. Dean, Neil J. Lant.

**Data curation:** John R. Dean, Neil J. Lant.

**Formal analysis:** Amber M. Cummins, Suzanne Lonsdale, John R. Dean, Neil J. Lant.

**Funding acquisition:** John R. Dean, Neil J. Lant.

**Investigation:** Amber M. Cummins, Adam K. Malekpour, Andrew J. Smith, Suzanne Lonsdale, Neil J. Lant.

**Methodology:** Amber M. Cummins, Adam K. Malekpour, Andrew J. Smith, Suzanne Lonsdale, John R. Dean, Neil J. Lant.

**Project administration:** John R. Dean, Neil J. Lant.

**Resources:** Suzanne Lonsdale, John R. Dean, Neil J. Lant.

**Supervision:** Andrew J. Smith, John R. Dean, Neil J. Lant.

**Validation:** Amber M. Cummins, Suzanne Lonsdale, John R. Dean, Neil J. Lant.

**Visualization:** John R. Dean, Neil J. Lant.

**Writing – original draft:** John R. Dean, Neil J. Lant.

**Writing – review & editing:** Amber M. Cummins, Adam K. Malekpour, Andrew J. Smith, Suzanne Lonsdale, John R. Dean, Neil J. Lant.

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
