## [Decision Letter · Decision Letter 0]

8 Mar 2023

PONE-D-23-02944Impact of vented and condenser tumble dryers on waterborne and airborne microfiber pollution.PLOS ONE

Dear Dr. Lant,

Thank you for submitting your manuscript to PLOS ONE. After careful consideration, we feel that it has merit but does not fully meet PLOS ONE’s publication criteria as it currently stands. Therefore, we invite you to submit a revised version of the manuscript that addresses the points raised during the review process.

We look forward to receiving your revised manuscript.

Kind regards,

Amitava Mukherjee, ME, Ph.D.

Academic Editor

PLOS ONE

“The authors thank the Master of the Worshipful Company of Launderers (2022), and its Education Committee, for funding in the form of a grant to AMC. Northumbria University is also thanked for provision of laboratory consumables. The authors are grateful to William Caufield of Procter & Gamble Newcastle Innovation Center for the microscopy of tumble dryer lint filters.”

“The study was partially funded by Procter

& Gamble Technical Centes Ltd, in the form of salaries for NJL, AKM and AJS and purchase of appliances and related laboratory consumables. In

addition to NJL, AKM and AJS, another member of Procter & Gamble staff contributed to the study as described in the acknowledgements but only the

co-authors were involved in the preparation of the manuscript. Procter & Gamble management gave approval to publish, but this process did not

influence the text of the manuscript.

All other expenses were paid by The Worshipful Company of Launderers, in the form of a grant to AMC, and by Northumbria University through the employment of JRD and SL and provision of consumables. The Worshipful Company of Launderers had no role in study design, data collection and analysis, decision to publish, or preparation of the manuscript.

https://www.launderers.co.uk/

There was no additional external funding

received for this study.”

Reviewers' comments:

Reviewer's Responses to Questions

**Comments to the Author**

1. Is the manuscript technically sound, and do the data support the conclusions?

Reviewer #1: Yes

2. Has the statistical analysis been performed appropriately and rigorously? 

Reviewer #1: Yes

3. Have the authors made all data underlying the findings in their manuscript fully available?

Reviewer #1: Yes

4. Is the manuscript presented in an intelligible fashion and written in standard English?

Reviewer #1: Yes

5. Review Comments to the Author

Reviewer #1: Impact of vented and condenser tumble dryers on waterborne and airborne microfiber pollution

Review Comments

Abstract:

The abstract seems to be bit lengthy. Can be shortened for better understanding

Introduction:

Line 85 – 86 – Inclusion of reference will be useful.

Though the necessity of the redesigning the dryer was addressed in the previous literature, the requirements of comparing vented and condenser dryers was not clear. How this previously studied Tumble driers are differed from the vented and condenser type tumble dryers ?

Line 130-145: This paragraph clearly reports that all the existing both vented, and condenser are designed to store their filtered fibers in it. However, their cleaning process and methods are dependent on the user decision.

Based on these pitfalls, study has taken two different objectives that did not address the issue discussed in line 130-145.

These issues can be included as an aim of the study

However, as author mentioned, the selected problems are addressed first time and their findings were need of the hour items in the research community.

Materials and methods

Line 195 – The protocols can be of some common standard like ISO or BS over a literature.

Line 264 – The mass measurements was provided however, the procedure did not refer any standards or previously published literature. Is it AATCC standard?? Can be clarified

Results and Discussions

The study provides an idea about the contribution of vented and condenser type dryer and their impact on the microfiber emission into wastewater, based on the type of cleaning and disposal of the filter or condensed water that a consumer adopts. This is the novelty of the study, and it is essential findings for the scientific community.

However, though the study cited the findings of the previous literatures that evaluated the normal tumble dryers emission, this study did not evaluate it, to compare with the considered to varieties. It is imperative to seek those findings, as the both vented and condenser types are the design variation of the abovementioned normal tumble drier. A comparison would have suggested how far the new designs are inferior or superior to the old one. This is a major draw back of the study. Due to these authors were not able to support their importance of the study over the previous literature like Kapp and miller and Kärkkäinen and Sillanpää.

Secondly, the authors did not compare the results of washing emission of the wash loads washed during the analysis. Inclusion of the washing data might be helpful for the readers to find the correlation or relationship or proportion of microfibers emitted in the corresponding washing and drying. It may also provide details including, whether the dryer emission is the really a pain point to address? While we did not have a proper commercial filter for the washing effluent as of now

Vacuum cleaning and disposal into a solid waste will directly contribute to the terrestrial microfiber pollution. That is also another problem which are equally important to the marine pollution but leatsly researched as of now. – This suggestion provided in Line 529 can be reconsidered before the publication of the manuscript.

Author should include their view point on this as it addresses the major issue

Addressing these points will improve the quality of the manuscript.

All the best.

6. PLOS authors have the option to publish the peer review history of their article (what does this mean?). If published, this will include your full peer review and any attached files.

Reviewer #1: **Yes: **Dr.R.Rathinamoorthy

---

## [Author Response · Author response to Decision Letter 0]

20 Mar 2023

Thank you for considering the above manuscript for publication in PLOS ONE. Attached is a revised version of the manuscript which I hope reflects the revisions requested in your letter dated 9th March 2023.

Point 1: I confirm that the manuscript meets PLOS ONE’s style requirements, including those for file naming.

Points 2 and 3: Regarding funding, we have removed mention of any funding information from the Acknowledgements or elsewhere in the manuscript, as requested. Furthermore, we have confirmed that the Worshipful Company of Launderers does not provide grant numbers with their awards. Below is a revised Funding Statement to reflect this.

The study was entirely funded by the following three sources:

The Worshipful Company of Launderers provided a grant to AMC through the Master (2022) of that institution and its Education Committee. No grant number was provided. The Worshipful Company of Launderers had no role in study design, data collection and analysis, decision to publish, or preparation of the manuscript.

https://www.launderers.co.uk/

Northumbria University funded the study through employment of JRD and SL, and provision of consumables. Only the co-authors affiliated to this institution were involved in study design, data collection and analysis, decision to publish, and preparation of the manuscript.

https://www.northumbria.ac.uk/

Procter & Gamble Technical Centres Ltd provided funding in the form of salaries for NJL, AKM and AJS and purchase of appliances and related laboratory consumables. In addition to NJL, AKM and AJS, another member of Procter & Gamble staff contributed to the study as described in the acknowledgements but only these individuals were involved in study design, data collection and analysis, decision to publish, and preparation of the manuscript. Procter & Gamble management gave approval to publish, but this process did not influence the text of the manuscript. 

https://www.pg.com/

Point 4: Details regarding ethical approval for the study have been added to the manuscript at line 182. 

Point 5: The reference list is complete and correct, and one reference has been added at the request of a reviewer (see below) compared to the original submission.

Reviewers’ comments

Abstract: This has been edited to sharpen the narrative and reduce its size to 240 words which is well below the recommended limit of 300 words. 

Introduction

Line 85-86: A reference has been included citing the legislation discussed.

The reviewer comments that “the requirements of comparing vented and condenser dryers was not clear. How this previously studied Tumble dryers are different from the vented and condenser tumble dryers”. We believe that this point is addressed in the Abstract and Introduction section “Contribution of tumble drying” where the previous work – which has all been focused on vented dryers, i.e. not involving condenser dryers, is summarized and the novelty of the present work clearly articulated in being the first to study condenser dryers and the first to study real consumer wash loads rather than new, clean, fabrics. 

Line 130-145: The reviewer commented that “existing both vented, and condenser are designed to store their filtered fibers in it. However, their cleaning process and methods are dependent on the user decision. Based on these pitfalls, study has taken two different objectives that did not address the issue discussed in line 130-145. These issues can be included as an aim of the study”.

I am interpreting this as a comment that the article contains measurements and discussion regarding disposal of fibers collected on the lint filter, which could result in waterborne pollution if consumers clean these components using water, but this is not set out a as a specific aim. The Introduction has been updated to explicitly build this into the aims.

Materials and methods

Line 195: The comment relates to whether common standards, e.g. ISO or BS, are relevant to the protocols. There are no such standards available for this type of research, although the methods employed have been successfully used in our previous articles (e.g. the two Lant et al. references) and articles cited in those.

Line 264: As described above (line 195) there is no AATCC method or other standard for this methodology.

Results and discussion

We found the general comments relating to this section rather confusing, especially the comment that “the study cited the findings of the previous literatures that evaluated the normal tumble dryers emission, this study did not evaluate it, to compare with the considered to varieties”. The present study comprehensively reviews and acknowledges the previous research in this area of microfiber pollution from tumble dryers. Those studies relate to airborne pollution from vented dryers. As condenser dryers are far more commercially important in other markets such as Europe where vented dryers are being phased out, this is the first study relating to condenser dryers, the first study to recognize that condenser dryers are a source of microfiber pollution from three sources, and a clear comparison is made versus vented dryers using consumer wash loads and the same test methodology. This novelty versus the earlier work of Kapp and Miller and others is clearly articulated in the abstract and introduction. It is not a case of ‘new designs’ (assumed to be condenser dryers) and ‘old designs’ (assumed to be vented dryers). Both types of dryer have been sold concurrently on the market for many years. As we have articulated, vented dryers cause airborne microfiber pollution (as described in earlier works) and may cause waterborne pollution depending on lint filter disposal method (present study) whereas condenser dryers have three potential sources of waterborne pollution – condensed water, cleaning of condenser, and cleaning of lint filter (if consumers wash this under water). The reviewer suggests that we should be including discussion of the relative superiority of ‘new’ (assumed to be condenser) versus ‘old’ (assumed to be vented) designs. We respectfully suggest that this is outside the scope of the study as that would involve weighing up the relative impact of causing air pollution (associated with vented dryers) or water pollution (mainly associated with condenser dryers). Our hope is that this article will bring the attention of the scientific community to the contribution of condenser dryers to microfiber pollution for the first time with an expectation that other groups may conduct further studies to either mitigate the issues or better understand the environmental impacts.

The reviewer also commented that there is no comparison with the microfiber release at the washing stage included in the present study to better understand the relative impact of washing and drying on microfiber pollution. We believe that this point is already addressed by Fig 8 which uses results from our previous comprehensive study of down-the-drain microfiber release from real consumer loads in washing machines (using similar appliances and washing conditions to those used in the present study) to make a holistic calculation of total microfiber release from all points during the washing and drying process. It is very clear from this data that when consumers dispose of dryer lint in municipal solid waste, that the washing stage will release more fibers to the environment than the tumble drying stage, although quantities released during drying are still very significant. However, if consumers follow the instructions of some dryer manufacturers and clean dryer lint filters in water, tumble drying becomes a more significant contributor than the washing stage. 

The reviewer’s final point is that disposal of collected microfibers in municipal solid waste will directly contribute to terrestrial microfiber pollution. This is an interesting argument and one that deserves further debate given that the main solution being implemented to mitigate microfiber pollution during laundering involves installation of filtration systems that involve disposal of the residue in municipal solid waste. We have updated the manuscript at the end of the Conclusions to highlight this point that disposal of microfibers in municipal solid waste could lead to some further environmental impacts that deserve further study.

Yours sincerely,

Dr Neil Lant, on behalf of the authors

lant.n@pg.com

R&D Senior Director / Research Fellow, Procter & Gamble

---

## [Decision Letter · Decision Letter 1]

12 Apr 2023

PONE-D-23-02944R1Impact of vented and condenser tumble dryers on waterborne and airborne microfiber pollution.

PLOS ONE

Dear Dr. Lant,

Thank you for submitting your manuscript to PLOS ONE. After careful consideration, we feel that it has merit but does not fully meet PLOS ONE’s publication criteria as it currently stands. Therefore, we invite you to submit a revised version of the manuscript that addresses the points raised during the review process.

We look forward to receiving your revised manuscript.

Kind regards,

Amitava Mukherjee, ME, Ph.D.

Academic Editor

PLOS ONE

Journal Requirements:

Reviewers' comments:

Reviewer's Responses to Questions

**Comments to the Author**

1. If the authors have adequately addressed your comments raised in a previous round of review and you feel that this manuscript is now acceptable for publication, you may indicate that here to bypass the “Comments to the Author” section, enter your conflict of interest statement in the “Confidential to Editor” section, and submit your "Accept" recommendation.

Reviewer #1: All comments have been addressed

Reviewer #2: All comments have been addressed

Reviewer #3: (No Response)

2. Is the manuscript technically sound, and do the data support the conclusions?

Reviewer #1: Yes

Reviewer #2: Yes

Reviewer #3: Yes

3. Has the statistical analysis been performed appropriately and rigorously? 

Reviewer #1: N/A

Reviewer #2: Yes

Reviewer #3: Yes

4. Have the authors made all data underlying the findings in their manuscript fully available?

Reviewer #1: Yes

Reviewer #2: Yes

Reviewer #3: Yes

5. Is the manuscript presented in an intelligible fashion and written in standard English?

Reviewer #1: Yes

Reviewer #2: Yes

Reviewer #3: Yes

6. Review Comments to the Author

Reviewer #1: I thank author for carefully addressing all the comments posted in the earlier review.

The manuscript can be accepted.

However, I would like to mention that for microfiber mass estimation recently AATCC has released the standard. Author can find the details over here: https://www.aatcc.org/tm212/

Reviewer #2: Overall, this edited version of the manuscript is thorough, well written and revised to incorporate previous reviewers comments. This article will be a valuable addition to the growing understanding of the contribution that clothes drying methods make to microfiber pollution.

There are just two minor revisions that must be made prior to publication. First, in lines 247-248, the figure being referred to is for Vented dryers, however the text is confusing. This must be changed to "Evaluation of the microfiber release from the vented dryer (Fig 3)" in Line 247. Second, line 498 should include "the pore size of lint filters in tumble dryers" for further clarity.

Reviewer #3: The manuscript entitled "Impact of vented and condenser tumble dryers on waterborne and airborne microfiber pollution" evaluated the release of microfibers from washing machines. The paper is well written and presents some thoughtful interpretations, however, no quantitative data/results were provided in the abstract proving "high levels of microfibers" (line 33) and "vented dryers were found to be significant contributors" (line 35).

Other comments include:

-QA/QC's were not provided in the paper. Did the researchers take precautions in the lab to reduce airborne microfiber contamination? (Lab coats, wipe down surfaces and equipment, etc)- If so, please add this in paper or supplementary

- Lines 311-315: Please provide the statistical software (plus reference) used in the Statistics section.

7. PLOS authors have the option to publish the peer review history of their article (what does this mean?). If published, this will include your full peer review and any attached files.

Reviewer #1: **Yes: **Dr.R.Rathinamoorthy

Reviewer #2: No

Reviewer #3: No

---

## [Author Response · Author response to Decision Letter 1]

21 Apr 2023

I thank Reviewer #1 for their comment that AATCC has released a standard method for the quantification of microfiber release. This is welcome progress, and I was involved in the early stages of its development. However, as it is an accelerated method, and limited in scope to microfiber release during clothes washing rather than drying, it was not suitable for the present study. 

Reviewer #2 requested two minor revisions to lines 247-248 and line 498 which have been made. The first is an important correction and we agree that the latter is a useful clarification.

Reviewer #3 commented that the abstract contains no data was included in the abstract to support the comments “high levels of microfibers” (line 33) and “vented dryers were found to be significant contributors” (line 35). The abstract has been updated to include all of these data.

Reviewer #3 asked about precautions to reduce airborne microfiber contamination. In addition to general good laboratory practice including wearing of laboratory coats, several references to additional precautions are mentioned in the manuscript including use of orange cotton and black polyester garments in line 182 (in order to easily differentiate these, and enable easy identification of any contaminant fibers), appliance cleanout procedures (lines 204 and 215) and use of a specialized laboratory with high levels of ventilation (line 230). 

Reviewer #3 also asked that we include details of statistical software in the Statistics section (line 311-315). This has been updated accordingly.

---

## [Editor Report · Decision Letter 2]

26 Apr 2023

Impact of vented and condenser tumble dryers on waterborne and airborne microfiber pollution.

PONE-D-23-02944R2

Dear Dr. Lant,

We’re pleased to inform you that your manuscript has been judged scientifically suitable for publication and will be formally accepted for publication once it meets all outstanding technical requirements.

Kind regards,

Amitava Mukherjee, ME, Ph.D.

Academic Editor

PLOS ONE
---

## [Editor Report · Acceptance letter]

2 May 2023

PONE-D-23-02944R2 

Impact of vented and condenser tumble dryers on waterborne and airborne microfiber pollution. 

Dear Dr. Lant:

I'm pleased to inform you that your manuscript has been deemed suitable for publication in PLOS ONE. Congratulations! Your manuscript is now with our production department. 

Kind regards, 

on behalf of

Professor Dr. Amitava Mukherjee 

Academic Editor

PLOS ONE